# Detrusor Underactivity in Men with Bladder Outlet Obstruction

**DOI:** 10.3390/biomedicines10112954

**Published:** 2022-11-17

**Authors:** Hsiang-Ying Lee, Chien-Sheng Wang, Yung-Shun Juan

**Affiliations:** 1Department of Urology, Kaohsiung Medical University Hospital, Kaohsiung 800-852, Taiwan; 2Department of Urology, School of Medicine, College of Medicine, Kaohsiung Medical University, Kaohsiung 800-852, Taiwan; 3Graduate Institute of Clinical Medicine, College of Medicine, Kaohsiung Medical University, Kaohsiung 800-852, Taiwan; 4Center for Cancer Research, Kaohsiung Medical University, Kaohsiung 800-852, Taiwan

**Keywords:** detrusor underactivity, bladder outlet obstruction, men

## Abstract

Detrusor underactivity (DU) and bladder outlet obstruction (BOO) are both common troublesome causes of lower urinary tract symptoms (LUTS) and often impact on quality of life simultaneously in men. This article aims to focus on DU with BOO in male patients. Methods: Original articles concerning DU with BOO were identified through literature research from PubMed and EMBASE database. We selected 38 articles in our review, including those concerning pathophysiology, evaluation, treatment and predictors for a successful BOO surgery for DU. Results: DU from BOO can result from several pathophysiological mechanisms. Although urodynamic study (UDS) is considered as a precise method to diagnose DU and BOO, there are some previous studies which proposed a non-invasive method to identify DU related to BOO. The treatment goal of DU is restoring bladder contractility using medication or surgery. Releasing outlet obstruction and resistance is the main strategy to restore bladder contractility when medication to directly increase bladder contractility has had limited efficacy. Conclusions: DU from BOO is poorly understood and is largely under-researched. The etiology and pathophysiology still need to be evaluated. Effective and safe medication to restore bladder contractility is also lacking. It remains valuable to perform further research to reveal the unknown aspects of DU.

## 1. Introduction

Detrusor underactivity (DU) and bladder outlet obstruction (BOO) are common causes of lower urinary tract symptoms (LUTS) in elderly men, with impact on quality of life [1]. DU was defined by the International Continence Society (ICS) as reduced strength and/or duration of detrusor contraction resulting in prolonged bladder emptying and/or incomplete bladder emptying within a normal time span on the basis of urodynamic testing [2]. The most common symptoms in men with DU are prolonged voiding duration with or without feeling of incomplete emptying, urinary hesitancy, hyposensitive bladder and weak streaming [3]. The incidence of DU and BOO increases with age, and the proportion of elderly men affected by DU is up to 48% (age > 70 years) [4]. The exact prevalence of DU in BOO men is difficult to evaluate because of the invasive multichannel urodynamic testing which may increase the incidence of associated morbidity during or after the examination. Urodynamic study (UDS) is the gold standard for diagnosing DU. It is impractical to perform invasive examination for all elderly men in the community [5,6]. 

DU and BOO may coexist and have similar symptoms including difficult voiding and incomplete bladder emptying, and elevated post-void residual (PVR) urine volume. BOO itself will also lead to DU over a long period of obstruction, which causes the development of detrusor hypertrophy, and DU is the main cause of decreasing bladder voiding efficiency (BVE) in men with BOO [7,8]. Clinically, UDS can identify the diagnosis of DU; however, it is difficult to determine the etiology of DU if it is from the result of long-term obstruction. Nevertheless, differentiating patients presenting with DU associated with BOO is important, because detrusor contractility impairment may be related to the efficacy outcomes of BOO treatments. Moreover, patients with DU may have some complications such as urinary tract stone formation, recurrent urinary tract infection and renal function impairment [2,9]. 

The definition of BOO by the ICS is characterized as elevated detrusor pressure with reduced urine flow rate due to obstruction during the voiding phase. The most common symptoms of LUTS related to BOO in elderly men result from benign prostate enlargement (BPE) or benign prostatic obstruction (BPO) induced by benign prostate hyperplasia (BPH) [10]. BPH with DU showed good short-term outcomes with BPH surgeries but it revealed more a prominent improvement of voiding in patients without DU [11,12]. Therefore, the rationale of routine preoperative UDS testing is still under debate. Preoperative UDS can identify patients with DU, which may affect surgical outcomes in BPH patients, so we can have more comprehensive treatment and follow-up strategies for patients undergoing surgery. Because DU with and without BOO has different characteristics, we aim to review up-to-date knowledge on DU with BOO which has not been comprehensively discussed before.

## 2. Evidence Acquisition

Literature research from the PubMed and EMBASE databases was conducted in November 2021, screening all topics about DU with BOO in men. The search strategy included the following keywords/mesh terms: “detrusor underactivity”, “underactive bladder”, “underactive detrusor function”, “hypotonic bladder”, “bladder outlet obstruction”, “benign prostate hyperplasia”, “men”. Subsequently, the searches were pooled with the limitation of men, adults and language (English); then, animal model studies and review articles were excluded. Congress abstracts and book chapters were also not included in the review. 

After removal of duplicates, two authors (HYL and CSW) independently screened the titles and abstracts in an electronic database to select the papers relevant to the review topic for inclusion. Only full-text articles written in English were included. Initially, a total of 144 articles, of which 80 were retrieved by searching Pubmed and 64 by searching EMBASE, were identified. After excluding duplicates and following evaluation of the abstracts of all articles, 81 articles were included. We further assessed full-text articles and excluded review articles and those irrelevant of our subjects. We finally included 38 articles for our review. A flow chart is shown in Figure 1.

Most of the series were retrospective, single-center studies. The sections we discussed included pathophysiological mechanisms, how to diagnose and evaluate DU, management of DU with BOO, and the impact of DU on BOO surgery. 

## 3. Pathophysiological Mechanisms

The various causes of underactive bladder were reported. It mostly involved myogenic failure, efferent or afferent nerve dysfunction and central nerve system (CNS) dysfunction [13,14]. Human interconnected detrusor muscle bundles extend from the bladder to the bladder neck and urethra as an integrated unit [15]. The impairment of bladder detrusor function caused by BOO has been inferred from animal study [16,17]. It is assumed that BOO will alter detrusor structure which becomes hypertrophic and hyperplastic to compensate for the high resistance of the bladder outlet, thus allowing maintenance of the voiding function. If patients do not receive surgical or medical treatment to relieve BOO, detrusor function will deteriorate to the decompensated stage after a prolonged period of BOO [18,19]. At that time, patients will suffer from more difficult bladder emptying that results from increased connective tissue deposition and the degree of muscle degeneration [20]. Due to the alteration of bladder structure, this seems to be an irreversible stage if there is persistent obstruction. Clinically, there is also an important issue regarding the timing of aggressive intervention in BOO. If we can relieve obstruction before the decompensated and irreversible stage, detrusor contractility can be recovered. 

Cho et al. investigated the impact of adenosine triphosphate (ATP) and nitric oxide (NO) in the urothelium of DU with BPH [21]. They included 30 patients with cold-cup biopsy from the mucosa of the bladder wall and divided them into DU and no DU groups. The results showed a significantly lower level of ATP in the DU group than in the no DU group. In contrast, there was no significant difference in endothelial nitric oxide synthase (eNOS) level between the two groups. The research assumed ATP was related to detrusor function and ATP level was positively correlated with bladder contractility index (BCI) and detrusor pressure. ATP in the urothelium may play a role in the pathophysiology of DU with BPH patients.

Jiang et al. demonstrated that DU in a BOO subgroup presents significantly higher expression of β3 adrenoreceptors and lower expression of iNOS. Similarly, their study also showed no significant different of eNOS level among comparison groups [22]. In addition, the expression of E-cadherin decreased in the DU group and was associated with severity of DU and urothelial dysfunction. β3 adrenoreceptors were considered to impact the bladder sensation; higher expression of β3 adrenoreceptors indicates decreased bladder sensation [23]. The decreasing activity of iNOS level is related to decompensated status after BOO, which contributed to the development of DU. Their research concluded that in male patients, BOO leads to urothelial dysfunction, suburothelial inflammation, cellular apoptosis and alterations in sensory proteins. 

Gheinani et al. [24] studied the role of microRNAs (miRNAs) in bladder contractility problems resulting from BOO using next-generation sequencing-derived transcriptome data. They found miRNA-regulated pathways are related to immune response, growth responses and apoptosis, which impact bladder hypertrophy and fibrosis. Some specific miRNAs and their mRNA targets were strongly regulated in BOO patients and can be used to understand the progress of bladder function from detrusor overactivity to DU status. The candidate miRNA may be used in new diagnostic and therapeutic tools used in deciding to make interventions earlier, to prevent BOO patients deteriorating to DU. 

Chronic bladder ischemia induced by BOO is also one of causes impairing bladder function. The discovery that excessive oxidative stress results from repeated ischemia/reperfusion cycles was confirmed by various animal models. It also induces bladder remodeling and eventually leads to decompensation status later [25,26,27]. Fusco et al. [26] proposed a three-stage model for BOO-induced bladder remodeling in humans which demonstrated multiple pathways involved in the process leading to bladder dysfunction. The pathologic change is from adaptive responses, including detrusor smooth muscle cell hypertrophy, reaching their limit with progressive extracellular matrix deposition, then finally urothelial dysfunction, and neuron and muscle cell degeneration. Furthermore, normal release of neurotransmitters for activation of ATP and purinergic (primarily P2 × 3) receptors present in suburothelial sensory nerves are impaired during ischemia, leading to impairment of neuromuscular control of the bladder [28,29]. 

A high percentage of neurologic deficits was detected by performing electrophysiology study in patients with DU and chronic urinary retention even without sacral neuropathy. Jiang et al. discovered a neurogenic impairment phenomenon including negative bulbocavernous reflex, urethral sphincter electromyography (EMG) reinnervation, decreased or absent EMG recruitment and decreased nerve conduction velocity amplitude in DU patients. The study also included patients with closed bladder neck while voiding. If the impact from BOO results in afferent or efferent neuropathy, surgery may be not satisfactory [30]. 

## 4. Diagnosis and Evaluation (Shown in Table 1)

Because DU and BOO have common symptoms, UDS is considered as the accepted method for the precise diagnosis of DU and BOO [31]. Nevertheless, uroflowmetry study is insufficient to differentiate between DU alone, BOO alone and BOO with DU although a study by Wada et al. stated that patients with DU alone have a higher incidence of sawtooth uroflowmetry pattern than BOO [32], and lower intravesical prostatic protrusion and BVE are also predictive factors for DU [33]. However, some previous research proposed that some non-invasive parameters, including specific symptoms and signs, can be differentially diagnosed in advance. Luo et al. [34] used non-invasive predictors to assess DU in BPH/LUTS men. Prostate volume and PVR are two significant predictors after multiple logistic regression analysis. After combining these two variables, the AUC was 0.774. They suggested clinical factors were beneficial to predict DU with BPH if patients cannot undergo invasive UDS.

**Table 1 biomedicines-10-02954-t001:** Enrolled studies correlate to findings with DU predictors in BOO patients.

Study	Design	Participants	Inclusion Criteria	Noninvasive Parameters as DU Predictors	UDS Parameters as DU Predictors
Luo et al. [17] (2017)	Retrospective study	704 men	BPH/LUTS patients who underwent urodynamic assessment	Smaller prostate volume (<30 mL), Higher post-void residual (PVR) (>400 mL)	
Nunzio et al. [18] (2020)	Retrospective analysis of a prospectively collected database	448 men	Aged 45 years or older with LUTS	Bladder wall thickness (BWT, mm): measured with suprapubic ultrasound, 3.5 MHz Qmax (mL/s) Nomogram	
Rademakers et al. [20] (2016)	Prospective trial	143 men	Treatment-naive men aged ≥40 years with uncomplicated LUTS	Detrusor wall thickness (DWT) ≤ 1.23 mm and bladder capacity >445 mL DWT: Anterior bladder wall with a 7.5 MHz ultrasound array and bladder filling ≥250 mL	
Gammie et al. [22] (2018)	Retrospective study	1612 men	Comparing 129 DU and 60 DU + BOO(DU: BCI < 100, BVE < 90%, BOOI < 20. DU + BOO: BCI<100, BVE < 90%, BOOI ≧ 40)	Higher average urine flow rate Less decreasing urinary stream More history of ≧1 urinary tract infection	Higher abdominal pressure at maximum flow rate (PabdQmax): straining
Oelke et al. [23] (2016)	Retrospective study	822 men	Aged ≧ 40 years with LUTs		WFmax-BOOI nomogram < 25th percentile groups Nomogram
Oelke et al. [24] (2014)	Retrospective study	786 men	Treatment-naive men aged ≥40 years with uncomplicated LUTS		BCI and WFmax rise with increasing BOO grade
Namitome et al. [25] (2020)	Retrospective study	909 men	Aged ≧ 50 years underwent pressure-flow studies	Older Small prostate volume Less urgency Weak streaming Low Qmax Nomogram	
Donkelaar et al. [26] (2017)	Retrospective study	1222 men	Aged ≧ 50 years underwent pressure-flow studies (Comparison LinPURR, BCI, WFmax)		LinPURR nomogram: simple and easy, no complex calculation BCI WFmax: more complex calculation Three methods have high correlation agreement
Guo et al. [28] (2017)	Retrospective study	67 men	Urinary retention men underwent UDS		Piso < 50 cmH2O: mechanical stop test BCI < 100
Tanabe et al. [29] (2011)	Retrospective study	288 men	Surgical indications underwent pressure-flow studies (Comparing operate and not operated with weak detrusor)		Schafer’s nomogram

BPH: Benign prostate hyperplasia, BOO: bladder outlet obstruction, BCI: bladder contractility index, BVE: bladder voiding efficiency, BOOI: bladder outlet obstruction index, DU: detrusor underactivity, LUTS: Lower urinary tract symptoms, LinPURR: linearized passive urethral resistance relation, Piso: isometric detrusor pressure, Qmax: Free uroflowmetry max flow, UDS: urodynamic study, WFmax: the maximum Watt factor.

Measuring bladder wall thickening (BWT) is another non-invasive method to evaluate DU. In a study by Nunzio et al. [35], they included male patients with LUTs and BPE to investigate non-invasive predictors for DU using clinical nomograms including BWT (mm), age (y) and free uroflowmetry maximum flow (Qmax, mL/s). These three variables presented a statistically significant difference for the prediction of DU in multivariate regression analysis. The study revealed a statistically significant median difference of 0.9 mm in BWT values between patients with bladder contractility index (BCI) < 100 and BCI > 100. To decrease variation and bias, the measurement of BWT was taken with the bladder filled to 150 mL with a 3.5 MHz convex probe which followed the principle of Manieri’s study [36]. However, this study only included small prostates (37.7 ± 24.4 mL) and mild or moderate LUTs. It needs further validation study to verify its clinical utility. Alternatively, Rademakers et al. [37] used ultrasound measurement of detrusor wall thickness (DWT) as a parameter to evaluate DU with LUTs. The results showed DWT ≦ 1.23 mm and >445 mL bladder capacity presented a higher possibility of DU. In contrast, Kalil et al. [31] pointed out that clinical variables including the International Prostate Symptom Score (IPSS) and PVR are not enough to differentiate patients with DU from those with BOO, and they suggest the need for combined UDS parameters for accurate diagnosis. 

Gammie et al. [38] compared groups with DU alone and DU with BOO, and discovered that some symptoms and signs have significant differences between the two groups. Their study showed less daytime micturition, larger maximum voided volume, higher abdominal pressure at maximum flow rate (PabdQmax) during UDS, higher urine flow rate, faster urinary stream and higher incidence of urinary tract infection in the group with DU alone. When patients presented with lower PabdQmax, it indicated a higher possibility of DU with BOO. Using PabdQmax combined with other signs and symptoms can be as a means to discriminate DU alone from DU with BOO.

Oelke et al. [39] developed a nomogram for assessment of bladder contractility by calculating the Watts factor (WF) and bladder outlet obstruction index (BOOI) as parameters. When patients are in the <25th percentile of the nomogram, it may refer to an association with DU. The same group also demonstrated that the BCI and the maximum Watts factor (WFmax) were associated with BOO grades. The higher BOO grades, the higher the BCI and WFmax values [40]. They suggested BCI < 100 or WFmax < 7 W/m^2^ may be considered as the threshold of DU. In the recent research by Namitome et al. [41], a prediction model for DU was developed for males with LUTs including BOO patients (65%). The included predictive variables were age, prostate size, IPSS and Qmax. It showed a higher possibility of pure DU when patients are older, have a smaller prostate size, experience less urgency and weak streaming, and have a lower Qmax which means they have higher nomogram scores. Donkelaar et al. [42] compared three methods to diagnose DU which only included 1222 men with LUTs. BOO patients were also enrolled. Schäfer pressure-flow nomogram, BCI and WFmax were used for grading, and pressure-flow parameters were necessary to calculate the level of linearized passive urethral resistance relation (LinPURR) in the Schäfer pressure-flow nomogram, BCI and WF. They demonstrated LinPURR and BCI contractility grading agreement are high in both obstructed and unobstructed men (97.2% and 97.7%, respectively). Because the calculation of WF is more complicated, LinPURR and BCI are good clinically relevant tools to evaluate detrusor contractility. It is important to evaluate detrusor contractility before prostate surgery because it is related to success rate and patient satisfaction. 

For men with DU and BOO who underwent surgery, only 64% of patients reported that they were satisfied with the surgery [12]. Guo et al. [43] found that a total of 60% men with urinary retention have BOO. This highlights that prostate surgery may not adequately control all urinary retention cases and also that not all urinary retention patients need receive prostate surgery. In addition to BCI, they also used isometric detrusor pressure (Piso) and detrusor reserve to diagnose DU. The definition of DU is Piso <50 cm H2O and detrusor reserve < 20 cm H2O. For men with overt urinary retention, Piso might be more suited to assessing the bladder strength. Nevertheless, performing prostate surgery for BPH patients with weak detrusor contractility is still feasible. Tanabe et al. [44] compared weak detrusor contractility patients who received prostate surgery and those who did not receive the operation. Patients who had the operation had more urinary retention episodes, less detrusor overactivity, a higher obstruction grade, and lower Qmax and voided volume. It is important to evaluate using pressure-flow study before surgery.

## 5. Management of DU with BOO 

Currently, there is no published consensus or set of guidelines on the management of patients with DU and the pharmacological or surgical treatments only have limited benefit. The goal of the treatment is to improve detrusor contractility, reduce outlet resistance, increase cortical perception of bladder sensation and reduce the complications arising from poor bladder emptying. In clinical settings, several factors need to be considered for determining the treatment including age, sex, etiology of DU, PVR, the presence of upper tract dilatation, and urinary tract infection.

Several pharmacological treatments attempting to improve detrusor contractility have been proposed. Many of them focus on bethanechol, which is a parasympathomimetic drug acting on muscarinic receptors (M_3_) and which showed only a limited effect with serious side-effects [45]. The role of parasympathomimetic agents in preventing or treating postoperative urine retention was discussed in a previous systematic review [46]. However, the evidence of parasympathomimetic agent effects on recovery of detrusor contractility is still not strong due to underpowered studies. Tomonori et al. showed that the combination of a cholinergic drug and an a-blocker is more effective than monotherapy for the treatment of voiding difficulty in patients with underactive detrusor. Patients may be unsatisfied with only cholinergic drug treatment because the effect of improving detrusor contractility is poor. In addition, there are cholinergic receptors over the urethra, so it may increase bladder outlet resistance when cholinergic drugs interact on the urethral receptors [47,48]. Transient receptor potential vanilloid channels (TRPVs) are another potential pathway to achieving bladder contraction. TRPV4 is a Ca2+-sensitive cation channel responsible for converting thermal energy to neural signals. TRPV4 agonist GSK1016790A are molecules that have been shown to increase bladder contractility [49]. Deruyver et al. investigated whether these molecules could increase bladder activity in a preclinical rodent model with DU and they found that it significantly increased the bladder contractility on cystometry [50]. Further study is needed before TRPV4 agonist can be used on humans. The efficacy of α1 -blocker and PDE5 inhibitor on DU patients have been studied by Yoshihisa et al. They discovered that after treatment with tadalafil (5 mg/day) or silodosin (8 mg/day) for 12 months, the maximum urinary flow rate and the mean bladder contractility index significantly improved [51].

To achieve proper bladder emptying and to reduce intravesical pressure, patients with DU may rely on clean intermittent catheterization (CIC), indwelling Foley catheter or cystostomy. Yet due to its impact on life quality, surgical treatment to decrease bladder outlet resistance is usually suggested to resume spontaneous voiding. Based on the obstruction site, the possible surgical options include urethral sphincter botulinum toxin A injection, transurethral incision of the bladder neck, and transurethral resection of the prostate (TURP). 

In DU patients with functional BOO such as spinal cord injury causing detrusor sphincter dyssynergia, peripheral neuropathy with detrusor areflexia, and dysfunctional voiding, botulinum toxin A can be injected to the urethral sphincter and the striated muscle will relax and the bladder outlet resistance may decrease [52]. Once the urethral resistance is reduced, patients can void more efficiently and their Qmax and PVR may also improve [53]. A prospective study conducted by Chen et al. has showed that after 1 month of 100 units of botulinum toxin A injection into the external sphincter, the previous DU or acontractile detrusor patients had increased Qmax and decreased residual urine, maximum urethral pressure, and detrusor leak point pressure [54]. Among those who could not void before the injection and relied on CIC, about 25% of them resumed spontaneous voiding.

Several studies have showed that patients with DU were able to regain spontaneous voiding after TURP. Dobberfuhl et al. reported that 79% of men with DU (BCI < 100) were able to void spontaneously after TURP and their mean Qmax and PVR were also improved significantly [55]. Blaivas et al. found similar results, i.e., that DU patients experienced improvements in American Urological Association Symptom Score, LUTS, Qmax and PVR after BPO surgery. None of the patients, who was CIC-dependent pre-operatively, needed CIC after the operation and all were able to void spontaneously [56]. Other therapeutic choices for DUA, including stem cells, regenerative therapy [57,58] and gene therapy [59], may be considered in the future but still need further comprehensive study to prove their efficacy. Due to the multifactorial causes of DUA, a combination approach may provide better recovery of voiding function [60].

## 6. The Efficacy of BOO Surgery for DU

Although TURP is still frequently used as a traditional therapy for BPH, lasers have been widely used in prostate surgery in the past two decades. Recent studies have also focused on laser surgery in the treatment of patients with DU. Holmium laser enucleation of the prostate (HoLEP) is used as a common surgical management technique for BPH. Its use in DU patients has been studied by Woo et al., and they reported that at the 6-month postoperative follow-up, patients in the holmium laser group demonstrated higher Qmax, lower PVR and greater improvement in the total IPSS when compared with the TURP group [61]. Choi et al. also reported their experience with high-performance system laser photoselective vaporization of the prostate (PVP) for the treatment of men with BPH and DU. They discovered that patients with BOO and DU had significant improvement in total IPSS, Quality of Life (QoL) score, Qmax and PVR at 1 month post operation and this remained significant at 12-month follow-up [62]. Similarly, after long-term following up to 50.9 months, the efficacy was still persistent and around 88.9% patients were free of catheter indwelling after HoLEP surgery [63]. A previous prospective trial also demonstrated significant improvements in LUTS and uroflowmetry results for hypocontractile patients after HoLEP. The improvements were not only in clinical symptoms, but in urodynamic parameters. Around 78.9% patients exhibited a significant return of bladder contractility. Therefore, most impaired detrusor contractility patients reported high satisfaction after receiving prostate surgery [64]. The recovery of detrusor contractility for patients with impaired detrusor contractility after TURP was also demonstrated by a study by Abdelhakim et al. It showed improving PdetQmax and BCI after 6-month serial follow-up [65].

However, this therapeutic effectiveness seems to decrease as the severity of DU increases [66]. After comparing BOO-only and BOO with the detrusor dysfunction group (including DO and DU), BOO-only patients showed better improvement in IPSS score after receiving HoLEP surgery. DU related to prolonged BOO may weaken the efficacy of prostate surgery. Therefore, early surgery intervention may be beneficial for preserving detrusor function. Yu et al. found that in the severe DU group (10 ≤ detrusor pressure at maximum flow rate (PdetQmax) < 20 cmH2O), the post-operative ineffective rate (PdetQmax < 40 cm H2O, Qmax < 10 mL/s, PVR > 50 mL, IPSS scored from 20 to 35 and QoL scored from 5 to 6) was as high as 52.9% [67]. Blaivas et al. also demonstrated that no matter what kind of endoscopic prostate surgery was undertaken, it revealed symptomatic improvements in patients with BOO and DU. However, an acontractile detrusor is a poor prognostic factor implicating a poor outcome [56]. 

## 7. Predictors for a Successful BOO Surgery for DU

Previous studies have shown that patients with DU and untreated BOO were able to benefit from BOO surgery [55,56]. However, a long-term urodynamic study of men with DU showed that resection of the prostate (TURP) does not always improve detrusor contractility, clinical symptoms and quality of life [68]. A study by Thomas et al. concluded that patients with obstruction and DU were able to void after surgery but no candidate preoperative urodynamic results can predict if patients can void or not [69]. Dobberfuhl et al. found that the ability of spontaneous voiding before surgery, Qmax, PdetQmax and BOOI > 40 were associated with increased odds of post operative spontaneous voiding. Preoperative PVR elevation was associated with post-operative failure to void [55]. Lee et al. reported that men with higher voiding detrusor pressure and larger bladder compliance had a better chance to regain spontaneous urination and better voiding efficiency. Moreover, 81.7% of DU patients undergoing transurethral prostate surgery could recover voiding efficiency of more than 50% [70]. The possible explanation is that patients with higher detrusor pressure might not lose their detrusor function completely and higher bladder compliance might indicate less bladder wall fibrosis. Wu et al. conducted a similar study and discovered that patients with higher detrusor pressure, greater voided volume and higher Qmax at baseline were associated with better voiding function recovery [71]. On the other hand, age, prostate size, baseline PVR, first sensation, full sensation and urgency sensation were unrelated to the voiding function recovery. The optimal cutoff value of these parameters has not yet been established but Chen et al. have proposed that bladder compliance of <80 mL/cmH2O may predict better bladder function recovery [72]. 

Zhong et al. [73] used two detrusor contractility parameters, BCI and WFmax, as predicting factors of the overall efficacy after TURP. The cut-off values for BCI and WFmax were set at 98.7 and 10.27 W/m2, respectively, which means the “low detrusor contractility” group is below 98.7 and 10.27 W/m2. After comparing the low detrusor contractility and high detrusor contractility groups, less LUTS improvement was shown in the low detrusor contractility group after receiving prostate surgery. Similarly, the low detrusor contractility group presented lower effectivity rates than the high detrusor contractility group. Plata et al. compared patients with DU and without DU undergoing PVP. Patients with DU had a higher incidence rate of temporary acute urinary retention after removing the urethral catheter postoperatively. Although IPSS score improvements were similar between groups, BOO with DU patients experienced less improvement in QoL [74].

In contrast, Cho et al. [75] evaluated the long-term efficacy of prostate surgery (HoLEP or PVP) both in patients with DU and without DU. Regardless of the presence or absence of DU, patients still had improving subtotal voiding and storage symptoms score, total IPSS, QoL index, Qmax, PVR and bladder voiding efficiency (BVE) up to three years after surgery. However, after dividing groups into mild and severe DU, both subjective and objective improvements in voiding symptoms, QoL, urine flow rate and BVE were presented less in severe DU patients than in the mild DU or without DU groups. Similarly, a previous long-term evaluation of efficacy after conventional TURP also demonstrated that improvements in total IPSS and QoL index can be maintained for up to 12 years irrespective of the presence or absence of DU [76]. The same research group of Cho et al. extended their observation of serial outcomes for up to five years. Generally, it showed sustained improvements in micturition symptoms, urinary flow rate, BVE and PVR after complete removal of the prostate using HoLEP [77].

It is still unclear how the detrusor contractility and the bladder function recover after BOO surgery in DU patients. The key is to identify whether the etiology of DU is due to prolonged BOO or not and this may impact the surgical outcome. Zhu et al. compared groups of patients with 20 ≤ BOOI < 40 and BOOI ≥ 40. Those with more severe outlet obstruction had greater improvement in subjective and objective symptoms after receiving TURP. In addition, the improvements decreased with an increasing DU degree in both groups [78].

## 8. Conclusions

There are various possible pathophysiological mechanisms related to DU which may influence the efficacy and choices of treatment. It is importance that treatment includes early intervention while the process of detrusor injury is still reversible. Although non-invasive methods are attractive tools for diagnosis of DU, including specified-symptoms-based study, uroflowmetry study and ultrasound measurement of DWT, urodynamic study is still the standard diagnostic examination. A combination of different parameters may increase the accuracy of diagnosis. Further possible comparison of the DWT and BCI indexes as well as LinPURR for the evaluation of the BOO and DU is warranted. Currently, there is no consensus or set of guidelines on the treatment for men with DU. Most of the studies contain both genders, mostly women, and the evidence may not be appliable to the male population. Due to the high prevalence of BOO in men, the surgical treatment for this population is usually focused on decreasing the bladder outlet resistance. Recent studies have shown that BOO surgery may be safe and effective in bladder function recovery and may help these patients regain spontaneous urination.

## Figures and Tables

**Figure 1 biomedicines-10-02954-f001:**
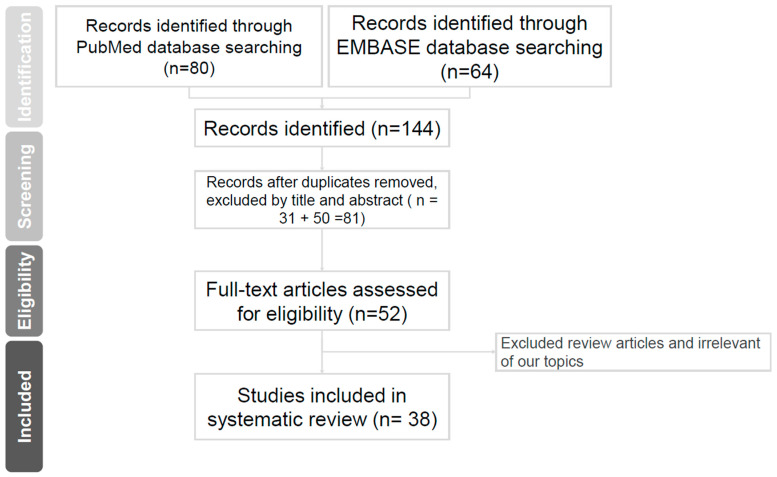
Evidence acquisition process.

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
