# Peer review of "Detrusor Underactivity in Men with Bladder Outlet Obstruction"

_biomedicines, 2022, doi:10.3390/biomedicines10112954_

Round 1

Reviewer 1 Report

The article entitled "Detrusor underactivity in men with bladder outlet obstruction 2" is a very comprehensive analysis of detrusor underactivity and bladder outlet obstruction. The introduction is correct. The authors should conduct systematic reviews according to Prisma systematic review guidelines, which should be highlighted in the article. Otherwise, the structure of the article is systematic. We agree with the need for a possible comparison of DWT and BCI indexes as well as LinPURR for the evaluation of the BOO. I suggest it should be highlighted more in the conclusion. 

Author Response

  • Thank you very much for your comments. This manuscript is a narrative review article, but I utilized the search strategy according to PRISMA from electronic databases for comprehensive review. I also have revised the conclusion as below.

Conclusion

There are various possible pathophysiological mechanisms related to DU which may influence the efficacy and choices of treatment. The importance of treatment is early intervention while the process of detrusor injury is still reversible. Although noninvasive methods are attractive tools for diagnosis of DU including specified symptoms-based, uroflowmetry study and ultrasound measurement of DWT, urodynamic study is still standard diagnostic examination. Combination of different parameters may increase the accuracy of diagnosis. Further possible comparison of DWT and BCI indexes as well as LinPURR for the evaluation of the BOO and DU is warrant. Currently, there is no consensus or guidelines on the treatment for men with DU. Most of the studies containing both genders, mostly women, and the evidence may not be appliable on male population. Due to the high prevalence of BOO in men, the surgical treatment for this population is usually focused on decreasing the bladder outlet resistance. Recent studies have shown that BOO surgery may be safe and effective in bladder function recovery and may help these patients regain spontaneous urination.

Reviewer 2 Report

The Authors carry out a narrative review on "Detrusor underactivity in men with bladder obstruction". The revision is effective however it would be useful to implement the paper with a separate sub-chapter in which histopathological aspects are treated separately.

Author Response

  • Thank you very much for your comments. This review article focused on DU from BOO in men; therefore, the search strategy only limited in human and men and the animal model experiments were excluded. Because the bladder histopathological aspects after relieving resistance most from animal studies, we did not include related paper. However, I also reviewed previous studies and organized them as below.

The efficacy of BOO reversal for DU from histopathological aspects

  Several previous animal studies found accumulation of oxidative stress with BOO. Using rabbit models, Lin et al. (1) discovered significant increase in oxidative stress biomarker levels including 8-hydroxy-2 ′-deoxyguanosine (8-OHdG), malondialdehyde (MDA) in the urine and plasma. The association between increased oxidative damage with obstructive bladder dysfunction was confirmed by another experiment. After reversing the obstruction status after 8 weeks, 8-OHdG and MDA returned back to control levels (2). Similarly, in another study, reversal of BOO is not only improving in the intravesical pressure and bladder compliance but in expression of cabonylation and nitrotyrosine which represent as potential oxidative stress products (3). The change of biomarkers may be considered as indicators of the severity of bladder dysfunction and the efficacy of a therapeutic intervention. Anti-oxidative products such as Gongjin-Dan (WSY-1075) and Eviprostate were investigated in BOO rat model. It showed these agents can decrease oxidative stress insults and prevent aggravating bladder dysfunction (4, 5).    

Reference:

  1. Lin WY, Chen CS, Wu SB, Lin YP, Levin RM, Wei YH. Oxidative stress biomarkers in urine and plasma of rabbits with partial bladder outlet obstruction. BJU Int. 2011;107(11):1839-43.
  2. Lin WY, Guven A, Juan YS, Neuman P, Whitbeck C, Chichester P et al. Free radical damage as a biomarker of bladder dysfunction after partial outlet obstruction and reversal. BJU Int. 2008;101(5):621-6.
  3. Lin WY, Wu SB, Lin YP, Chang PJ, Levin RM, Wei YH. Reversing bladder outlet obstruction attenuates systemic and tissue oxidative stress. BJU Int. 2012;110(8):1208-13.
  4. Jung JW, Jeon SH, Bae WJ, Kim SJ, Chung MS, Yoon BI et al. Suppression of Oxidative Stress of Modified Gongjin-Dan (WSY-1075) in Detrusor Underactivity Rat Model Bladder Outlet Induced by Obstruction. Chin J Integr Med. 2018;24(9):670-675.
  5. Oka M, Fukui T, Ueda M, Tagaya M, Oyama T, Tanaka M. Suppression of bladder oxidative stress and inflammation by a phytotherapeutic agent in a rat model of partial bladder outlet obstruction. J Urol. 2009;182(1):382-90.

Round 2

Reviewer 2 Report

the paper can be published after a minor revision to correct the linguistic style